# High-yield parallel fabrication of quantum-dot monolayer single-electron devices displaying Coulomb staircase, contacted by graphene

Joel M. Fruhman [1✉], Hippolyte P.A.G. Astier[1✉], Bruno Ehrler [1], Marcus L. Böhm[1], Lissa F. L. Eyre [1], Piran R. Kidambi[2,3], Ugo Sassi[4], Domenico De Fazio [4], Jonathan P. Griffiths[1], Alexander J. Robson [5], Benjamin J. Robinson [5], Stephan Hofmann[2], Andrea C. Ferrari [4] & Christopher J. B. Ford [1✉]

It is challenging for conventional top-down lithography to fabricate reproducible devices very close to atomic dimensions, whereas identical molecules and very similar nanoparticles can be made bottom-up in large quantities, and can be self-assembled on surfaces. The challenge is to fabricate electrical contacts to many such small objects at the same time, so that nanocrystals and molecules can be incorporated into conventional integrated circuits. Here, we report a scalable method for contacting a self-assembled monolayer of nanoparticles with a single layer of graphene. This produces single-electron effects, in the form of a Coulomb staircase, with a yield of $87 \pm 13\%$ in device areas ranging from $< 800\,\mathrm{nm}^2$ to $16\,\mu\mathrm{m}^2$, containing up to 650,000 nanoparticles. Our technique offers scalable assembly of ultra-high densities of functional particles or molecules that could be used in electronic integrated circuits, as memories, switches, sensors or thermoelectric generators.

[1] Cavendish Laboratory, University of Cambridge, Cambridge, UK. [2] Department of Engineering, University of Cambridge, Cambridge, UK. [3] Department of Chemical and Biomolecular Engineering, Vanderbilt University, Nashville, TN, USA. [4] Cambridge Graphene Centre, Cambridge, UK. [5] Department of Physics, Lancaster University, Lancaster, UK. ✉email: joel.fruhman@cantab.net; hipp.astier@cantab.net; cjbf@cam.ac.uk

The assembly of molecules and nanoparticles (NPs) into nanoscale circuitry is a potential route to device miniaturisation[1] and, as such, could help maintain the exponential increase in component densities. Furthermore, single-electron transport (SET), as realised in this way, can produce novel electronic behaviours that are desirable in commercial systems, such as negative differential resistance[2,3]. Such transport was demonstrated in scanning-tunnelling-microscope experiments on granular films[4], which progressed to lithographically defined quantum dots (QDs) with adjustable tunnel barriers[5]. Separately, tunable barrier gates were added with an extra 'plunger' gate to control the number of electrons without changing the tunnelling probability[6]. These dots are too large to work much above 1 K, so self-assembled, nm-sized clusters[7] and single-molecule junctions[8,9] were investigated using scanning-probe techniques. However, the severe difficulty of contacting such small objects with a wafer-scale process has thus far prevented the integration of nanocrystals and molecules into conventional microelectronic circuitry.

The variety of electronic behaviour in organic molecules and NPs may enhance Si-based devices[10]: molecules or NPs used as functional components in microelectronics offer smaller, faster, more energy-efficient electronic and photonic systems. Single-molecule and single-NP junctions usually display much variability in their electrical responses[11], owing to the many atomic-scale configurations available to them. The use of a self-assembled monolayer (SAM) of identical molecules allows one to average the variation due to the attachments of the molecules and hence arrive at repeatable electronic behaviour[12]. In addition, self-assembly can take place on Pt, which is compatible with CMOS processes. Whilst NPs have the advantage of being slightly bigger than molecules, and so are easier to contact or observe, their ~ 10% size variation can cause desirable electronic responses, such as Coulomb staircase, to be washed out in SAMs. Thus, most research studying these behaviours has addressed individual NPs, which is incompatible with mass-fabrication.

Nanogap junctions containing SAMs were fabricated using shadow evaporation[13–15], mechanically controlled break-junctions[16–19], electromigration[20] and NP chains[21]. Other approaches focus on nanopores[22], cross-wires[23], direct metal transfer[24], vacuum spray[25], eutectic Gallium-Indium (EGaIn) liquid metal[26] and vertical growth[27,28]. Whilst these processes successfully probe zero-dimensional electronic structure, each either measures the averaged electronic behaviour of many NPs/molecules or sacrifices potential parallel fabrication to measure just single-digit numbers of NPs/molecules. These are termed the "ensemble-molecule" and "single-molecule" regimes, respectively[29].

Graphene as an electrode has been explored for both regimes. Such studies include scanning-tunnelling-microscope measurements of alkanedithiol molecules with single-layer graphene (SLG) as a bottom electrode[30], laterally spaced SLG nanogaps[31–33], and using graphene[34,35], graphene oxide films[36] and EGaIn[37] as top electrodes to contact SAMs. The electronic properties of a SLG-sandwiched CdSe nanocrystal heterostructure have also been measured[38]. However, all of this research, so far, reports either the average of a large number of varied contributions, which is scalable, or single-molecule/NP behaviour, which is not.

Here, we present a SLG-covered SAM of NPs that produces functioning SET devices displaying a Coulomb staircase in their $I–V$ characteristics with a yield of $87 \pm 13\%$. The fabrication uses ensemble techniques, such as self-assembly, and layer-by-layer lithography, both of which are scalable. The devices work at temperatures up to at least 70 K and demonstrate unaltered electronic behaviour after a year stored in air at room temperature.

## Results

**Device structure, fabrication and characterisation.** Each device contains an array of double-barrier NP-molecule structures in parallel contacted between common source and drain electrodes (Fig. 1a,b,g). These are created by sandwiching a single layer of semiconducting PbS QDs between Au and SLG (grown by chemical vapour deposition, CVD, see Methods). The QDs are capped with insulating ligands and bonded to an alkanethiol molecule that is itself part of a molecular SAM assembled on the Au. This results in Au/tunnel barrier/QD/tunnel barrier/SLG junctions, thousands of which in parallel make up a device. SLG's thinness, strength, flexibility, high electrical and thermal conductivity, impermeability to gases and ability to sustain large current densities[39,40] ensure good electrical contact without the risk of shorting through or damaging the films, as is typically seen with top metal electrodes[41,42]. Moreover, CVD can produce wafer-scale sheets of SLG, allowing for scalable device fabrication[43–45].

The procedure described in Methods produces ~1400 devices in each batch. We use a 1,6-hexanedithiol SAM (C6S2) to attach preformed colloidal oleic-acid-capped PbS QDs[46] to an Au electrode and cover the resulting SAM with SLG (Fig. 4). ~10% of our devices short electrically, and $87 \pm 13\%$, with areas less $< 2\,\mu m^2$, display Coulomb staircase[47,48] (see Fig. 1e and its caption) when a voltage $V$ is applied across the device (Fig. 2).

We first consider the fabrication variables that best predict the occurrence of Coulomb staircase, then we fit ~ 1000 $I–V$ steps to gather step parameters and quantify the step variation within and across devices. We also perform combined ultrasonic force microscopy and atomic force microscopy (AFM), scanning electron microscopy (SEM) and transmission electron microscopy (TEM) to gain further structural information. All of this is used to explain why we see a consistently high yield of SET characteristics for device areas ranging from $<800\,nm^2$ up to $16\,\mu m^2$.

Devices are fabricated on a $SiO_2$-coated Si substrate in a $160\,\mu m \times 30\,\mu m$ rectangular region at the centre of each sample (Fig. 1b, f). In this area, 39 devices are made simultaneously by patterning the SLG such that it only covers the tips of the 39 QD-coated Au electrodes (seen as vertical lines in Fig. 1b). Each fabrication cycle (batch) produces 36 samples (~1400 devices) but this can be scaled up to much larger numbers by increasing the wafer size. The SLG also makes direct electrical contact with three horizontal electrodes, positioned in the centre of the devices, that are left bare. In all other areas, the SLG lies on the $SiO_2$.

By staggering the Au electrodes such that they span the possible positions of the etched SLG edge—produced by optical lithography (OL) misalignment (Fig. 1g)—a range of device areas can be obtained for a given sample that, in some cases, gives smaller areas than could have been produced with a non-stochastic approach, the goal being to minimise these areas and observe the most interesting electronic behaviour. In these OL samples, device areas range from $0.18 \pm 0.16$ to $18.3 \pm 0.2\,\mu m^2$. In one batch, additional electron-beam lithography (EBL) reduces areas further by up to a factor of 2500, resulting in samples with areas from $<0.0025$ to $2.15\,\mu m^2$ (Fig. 1h).

**Measurements and types of results observed.** The devices are measured at 4.2 K. $I–V$ curves are obtained with triangle bias sweeps that are repeated and slowly increased in magnitude until each device's voltage limit is reached (see Methods).

Based on the $I–V$ measurements, the devices are classified into five categories, each representing a distinct electronic behaviour: (1) repeated current plateaux (Fig. 2) (labelled 'step curves', StC), (2) no current plateaux and non-Ohmic conduction of a type normally seen in tunnel-barrier junctions[26], where current is

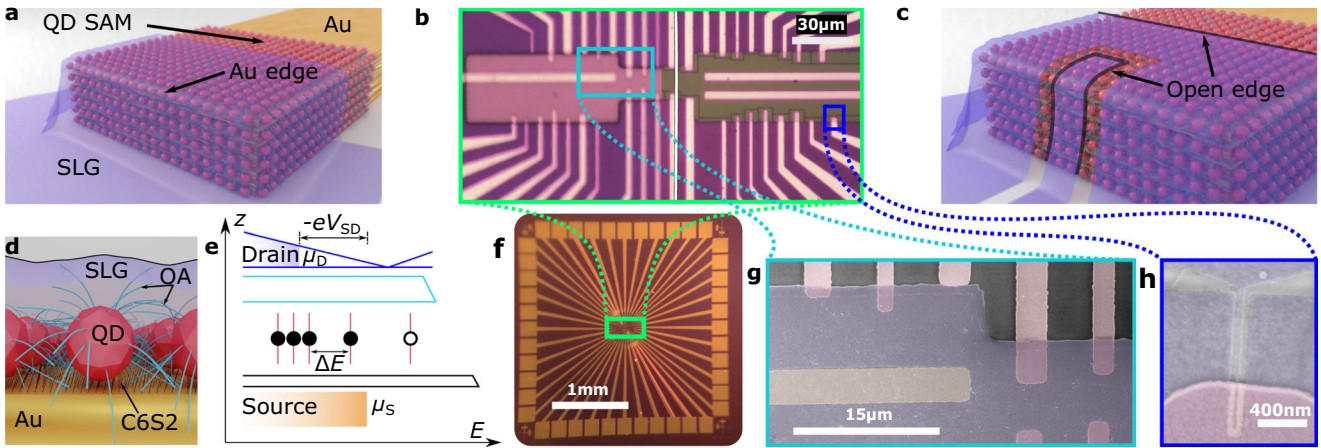

**Fig. 1 Device Structure. a** Single device: a 1,6-hexanedithiol (C6S2) SAM is bonded to evaporated Au fingers, and quantum dots (QDs) capped with oleic acid (OA) are then assembled on the SAM. SLG is draped over the entire structure to form a contacted array of double-barrier junctions in parallel. **b** Central device region of a single sample containing 39 individually addressable devices with different areas. Devices are created where SLG overlaps the tips of the Au electrodes (vertical lines). The left of the image shows an OL sample and the right an EBL one, where device areas are reduced further by EBL. The central horizontal electrodes are left bare for contacting the SLG. **c** Computer render of an EBL device showing its open (etched) edge highlighted in the foreground. In the background, the open edge of the original OL device. The open-edge length:area ratio of a device increases when EBL is used to reduce its area. **d** Single double-barrier structure (centre) as part of a larger array. C6S2 and OA form tunnel barriers between the QD and Au and SLG, respectively. **e** Energy structure of a junction aligned with the picture in (**d**). The Au and SLG electrodes are the source and drain, respectively. The C6S2 SAM and the QD's OA ligands form tunnel barriers at the Au/QD and QD/SLG interfaces, respectively. The QD has a discrete energy spectrum due to its addition energy $E_C = e^2/2C$, where $C$ is the QD's capacitance, and its single-electron energy levels. In this example a negative source-drain bias $V_{SD}$ is applied across the junction. This determines the energy window directing the transport by altering the electrodes' electrochemical potentials. The available energy $-eV_{SD}$ has just increased enough to allow a second electron to reside in the QD. The electrons do not tunnel out quickly as the exit barrier is thicker than the entrance one, but the total probability of leaving becomes twice what it was with one electron, and hence the tunnelling current steps up by about a factor of two. Addition of extra electrons (or holes) like this is exhibited as steps in the $I - V$ characteristic, dubbed a Coulomb staircase. **f** Single sample showing each device in the central region as individually addressable via bond pads on its perimeter. **g** False-colour SEM image of 7 OL devices (electrodes shown vertically in pink), each with a different area ($0.34 \pm 0.22 - 11.9 \pm 1.2\,\mu m^2$) arising as a result of the electrodes' relative positions spanning the optical-lithography alignment accuracy of the SLG patterning (shown in blue). A horizontal grounding electrode with no SAM is seen at the left of the image in yellow. **h** False-colour SEM of single EBL device (like that in **c**) with area $11500 \pm 2300\,nm^2$.

suppressed at low voltage, then increases rapidly at high voltage (see Supplementary Fig. 9b) ('smooth curves', SmC), (3) linear characteristics (see Supplementary Fig. 9a) ('short circuits', ShC), (4) no conduction ('open circuits'), (5) junctions exhibiting erratic conduction above some threshold voltage ('breakdown curves'). 4, 5 occur in <8% of the devices and are ignored as they likely result from random processing failures unrelated to the junction.

StC arise from QD Coulomb blockade and ShC from direct contact between SLG and Au electrodes (ShC have comparable resistances to ground contacts and devices in which the SLG is deliberately in direct contact with the Au: 1–730 kΩ with a mean ~54 kΩ). SmC are likely to be conduction through just the alkanethiol SAM because their shape and conductivities match control devices containing the C6S2 SAM only, without QDs, in which the junction structures take the form Au/C6S2/SLG (see Supplementary Fig. 9d). However, the SmC conductivity range overlaps that of the StC for both EBL and OL devices, implying that some of these SmC may arise from the blurring out of sets of steps from parallel QD double-barrier junctions (see Supplementary Fig. 9c).

### Coulomb-staircase curves

A series of reproducible discrete steps in current as a function of source-drain voltage (Fig. 2) is a signature of SET and is termed Coulomb staircase[47,48]. Each current step occurs when the increasing bias enables one more charge carrier to occupy the QD, providing a step change in the number of states available for tunnelling out of the QD, and hence in the

probability of tunnelling through it (Fig. 1e). Whilst Coulomb blockade can be observed when series tunnel barriers are approximately equal, Coulomb staircase requires asymmetric barriers so that multiple electrons or holes can accumulate in the QD, with low probability of tunnelling out and high probability of being replenished if they do. In our devices, the tightly packed C6S2 SAM provides a fixed covalently bonded barrier ~0.83 nm long between the Au and QDs (see Methods). The oleic acid molecules that coat the QDs provide the other tunnel barrier in the junction, bridging SLG and QDs via van-der-Waals bonding. As this barrier length is ~1–2 nm (see Methods), the asymmetric-barrier condition should usually be fulfilled.

Multivariate logistic regression is used to show that device area predicts the occurrence of both StC and ShC with a $p$ value of 0.00177 (Fig. 3e) (see Supplementary Note 1 for further discussion).

Whilst grouping of the StC measurements seems justified based on curve type, there is a significant variation of electronic behaviour within the group. Step heights ($\Delta I$) and the voltage increase required to induce successive current steps ($\Delta V$) are not constant between or within devices (see Supplementary Figs. 3 and 5 and Supplementary Note 2). However, almost all Coulomb-staircase devices display a very high level of electronic stability. Some are swept hundreds of times and remain unaltered, save for minor lateral shifts in their traces (see Supplementary 6, 7 and Supplementary Note 2). After thermal cycling, i.e. warming devices up to room temperature and then cooling them back to 4K, devices often retain the electronic structure with only minor

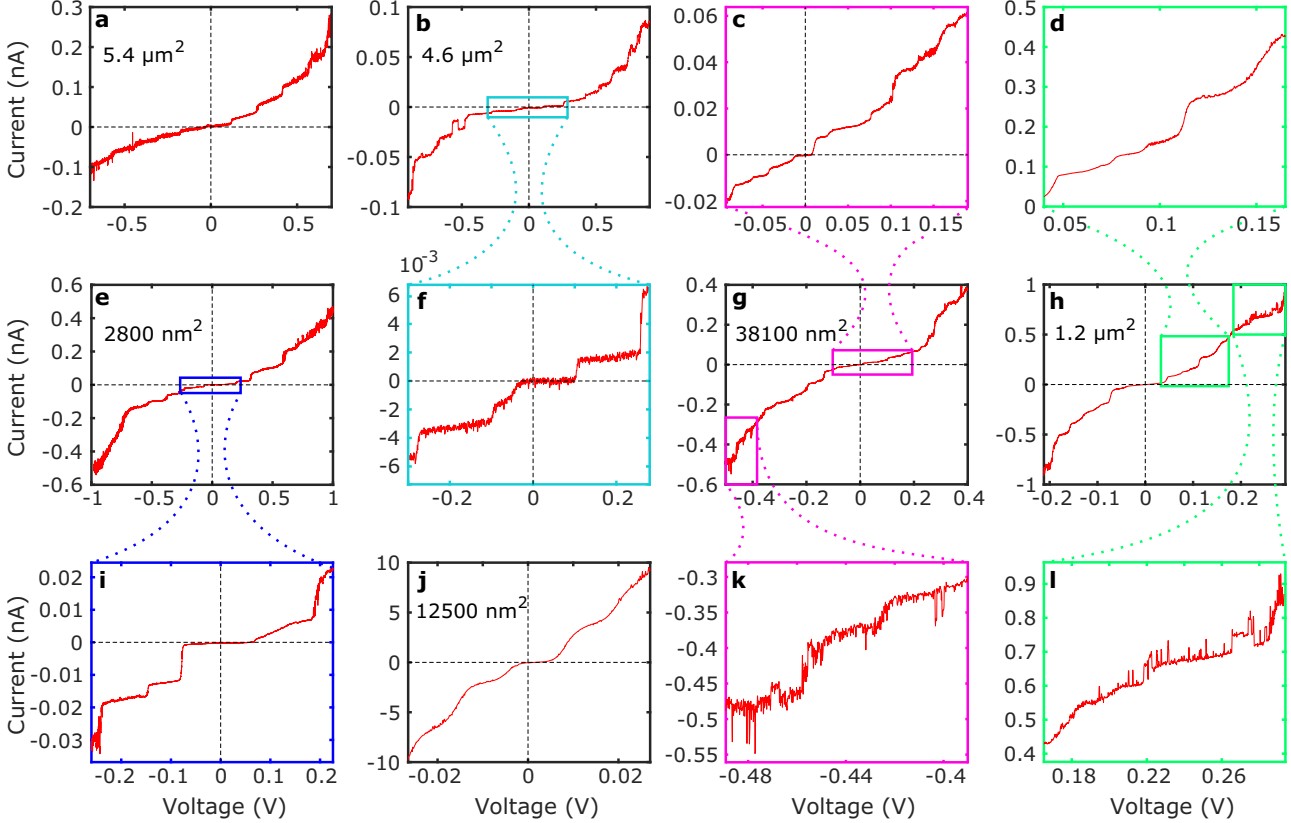

**Fig. 2 Devices made using optical (OL) or electron-beam (EBL) lithography, displaying Coulomb-staircase behaviour ('step curves', StC).** The sweep direction is reversed when each device reaches a voltage where its staircase becomes unresolvable, so the voltage range never exceeds ±1 V. The step width $\Delta V$ and height $\Delta I$ vary between and within devices. All sweeps show random telegraph noise (RTN) and resistances ~ MΩ. The selected sweeps are representative of the dataset. Devices show a high degree of stability over up to 100 sweeps. **a** I–V measurement showing a StC in an OL device with a ~5.4 $\mu m^2$ area. The differential conductance at 0 V ($G_0$) is $1.1 \times 10^{-9}$ S/$\mu m^2$ and the device shows distinct current plateaux in the range $|V| < 1$ V. This behaviour is stable: in > 20 sweeps only minor lateral shifts are observed. **b, f** A 4.6 $\mu m^2$ EBL device with StC and a variety of $\Delta V$ across the trace and increasing $\Delta I$ with increasing V. The device has $G_0 = 4.1 \times 10^{-10}$ S/$\mu m^2$. **f** This junction's smallest steps around the origin. The device is stable over >100 sweeps, except for some changes in the RTN height around − 0.6 V. **c, g, k** A 38100 $nm^2$ EBL device showing a wide range of $\Delta I$. The largest step height is 45 times bigger than the smallest. The device is stable for >50 sweeps and has $G_0 = 7.8 \times 10^{-10}$ S/$\mu m^2$. **c** Smaller step structures around the origin. **k** RTN at low voltage. **d, h, l** A 1.2 $\mu m^2$ EBL device with $G_0 = 8.4 \times 10^{-10}$ S/$\mu m^2$. **d** Shows the finer step structures close to the origin that span one order of magnitude. Low-frequency RTN in (**l**) is likely a result of trapped charges in the QDs' surface states or electron excitations inside the QDs. **e, i** A 2800 $nm^2$ OL device with ten steep steps in the $|V| < 1$ V range. The device has $G_0 = 4.7 \times 10^{-10}$ S/$\mu m^2$ and repeatable behaviour for >50 sweeps. **i** The first negative-bias step has a larger $\Delta I$ than the second negative step that follows as the voltage is made more negative. **j** A 12500 $nm^2$ EBL device with a high $G_0$ ~ $2.8 \times 10^{-8}$ S/$\mu m^2$. This device's higher conductance combined with its smoother current steps suggests multiple QDs conducting in parallel, all contributing some current to each step.

changes to the sizes of some curve features. The lack of periodicity in $\Delta V$, combined with the variations described (see Supplementary Note 2), indicate multiple QDs conducting in parallel. Furthermore, the inhomogeneity of behaviours between devices implies that the double-barrier structures are not identical, consistent with the TEM in Fig. 4a–c.

## Discussion

Our measurements indicate that StC are a consequence of currents through individual QDs superposing in such a way that they do not mask one another. As the number of parallel conduction pathways becomes large, steps in the I–V characteristic may be washed out, which could cause QD transport to be mislabelled as SmC, but the similarity between the shapes of SmC and of the control set makes this unlikely. Washed-out StC are more likely to appear as low-quality staircases with faint undulations, as occasionally observed.

Since the clear Coulomb staircase in many devices must result from most QDs *not* contributing significantly to the current, we propose three mechanisms for reducing the number of active QDs. (1) The electron tunnelling rate through a thick barrier varies exponentially with its thickness. AFM imaging shows an Au surface roughness $R_q = 1.1$ nm and $R_a = 0.82$ nm, where $R_q$ and $R_a$ are the one-dimensional root-mean-square and arithmetic-mean roughness respectively (see Supplementary Fig. 8). TEM shows a normally distributed QD size range with mean 5.0 nm and standard deviation 0.8 nm (Fig. 4a–c). Such a spread, combined with SLG's ability to remain suspended over micron-sized gaps[50], results in a range of QD-SLG tunnel-barrier lengths[51,52] (see Supplementary Note 3). The exponential dependence, combined with the variation in barrier lengths, reduces the number of QDs contributing to a device's current. (2) Quartz crystal microbalance (QCM) experiments on the monolayer (Fig. 4e, f) and AFM imaging (Fig. 4i) show gaps between the QDs ~ 9 nm on average. This further reduces the number of

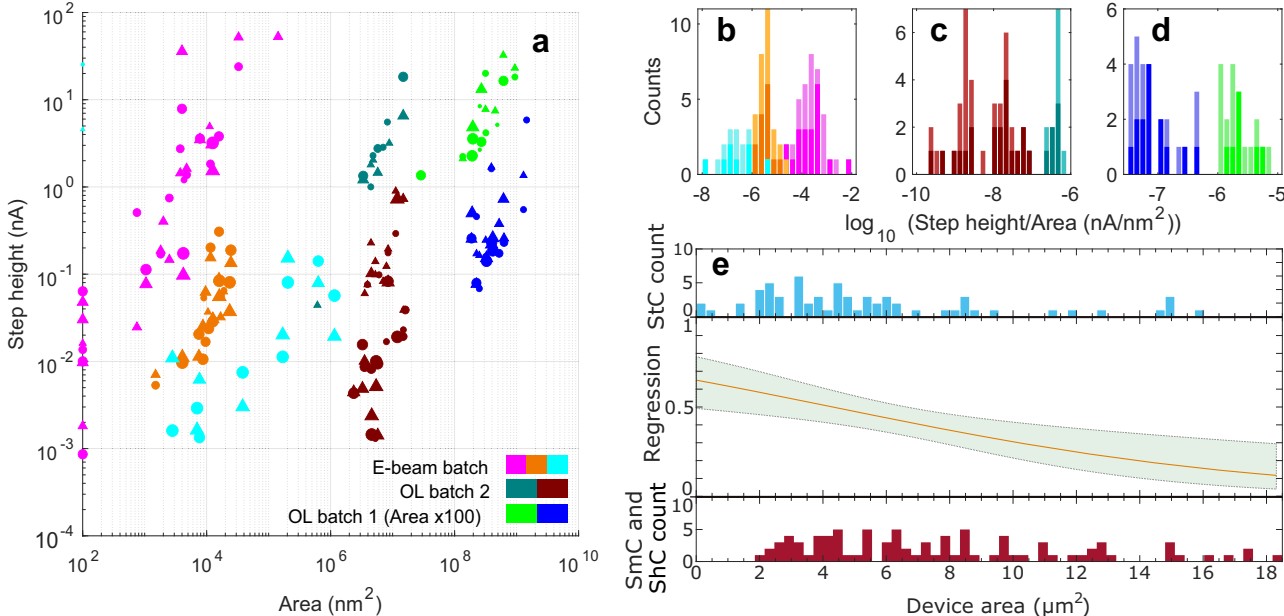

**Fig. 3 Correlation of step height and probability with area. a** Current heights of the first positive-bias (circles) and negative-bias (triangles) steps vs device area, for all StC. The size of each data point corresponds to the quality of the I–V trace: larger points represent higher-quality StC with more steps, less noise and/or sharper steps. Devices with areas <100 nm$^2$ are shown at 100 nm$^2$. OL batch 1 is offset laterally by a factor of 100 for clarity. The data cluster into groups and show a positive correlation when devices are split into 36-sample fabrication batches. Within each batch there is an additional separation into discrete conduction groups, each displaying a different range of current step heights. **b–d** Histograms of step height divided by area for each batch in (**a**), with matching colours, showing that the distributions have widths of at most an order of magnitude. **e** Logistic-regression analysis estimating the probability of StCs for different OL areas. Various independent variables have been tested, including types of device perimeter, but device area is the best predictor of StC and ShC with a p value of 0.00177. In the analysis, StC (SmC and ShC) are represented as 1 (0). Green banding around the probability line shows the 95% confidence interval across the area range. Histograms above and below the regression line show the curve-type counts for different device areas. Reducing the device area from 18 to <1 $\mu$m$^2$ increases the probability of measuring a StC from 0.12 to 0.65.

QDs contributing to a device's current, as SLG can bridge these gaps. (3) Water molecules or contamination during fabrication could lower the number of active QDs. Any dust or polymer resting on the assembly will render the QDs beneath inactive.

To further understand these data, hyperbolic tangent functions are fitted to all individual steps across all StC (see Supplementary Note 4). For the first positive and negative steps (those closest to the origin), $\Delta I$ increases as a function of device area (Fig. 3a–d). Since, as discussed above, the step is unlikely to include current contributions from many QDs, it is probably the result of larger areas containing QDs with smaller effective barrier lengths—QDs in the tails of this length distribution are more likely to be present in large-area devices. This correlation is clear when the data are separated by fabrication batch (collections of 36 samples) (see Supplementary Note 5).

The trends in $\Delta I$ vs area for each batch can be split into groups showing higher and lower step heights. These form distinct clusters in the plots and appear in all batches (see Supplementary Note 5). One possible explanation is that sometimes QDs can be pulled into fixed positions through the C6S2 SAM below, when under bias, making more direct contact with Au. This would reduce the effective barrier length of those junctions and increase $\Delta I$. The problem with such an explanation is that it must be a consequence of some physical event, e.g. a less dense or thinner SAM region, itself associated with some finite probability per unit area of occurring. Thus, the high-conductance trend should appear more often in larger devices, though currently there is no sign of this.

The trapped-charge effects point to local electronic behaviour in the SAM, hence corroborating the claim that our procedure allows for single or low-number QD measurements in large-area devices. In addition, there is a correlation between differential

conductance at zero bias ($G_0$) and current step height $\Delta I$ (or the ratio of step height to the voltage at which the step occurs) (see Supplementary Fig. 2). Since $G_0$ is a global property of a single device, which results from all contributions from every part of a device, and a current step corresponds to electrons in a single QD overcoming Coulomb blockade, this correlation means that these single QDs must provide most of the conductance at $V = 0$. A similar argument can be used when looking at the relation between $G_0$ and the current and voltage values just before the first positive and negative steps ($V_s$, $I_s$): $G_0$ correlates well with $I_s/V_s$ (see Supplementary Fig. 2c). Furthermore, the fact that all qualities of StC fit into this pattern, even when there is only one step in the curve, is evidence that all devices behave similarly and show tunnelling through low numbers of QDs.

In summary, our hybrid technique combines top-down lithography with bottom-up NP/molecular assembly to produce wafer-scale compatible devices that reliably display single-electron effects. These devices do not require nanoscale electrodes or nanogaps to make contact with single or low numbers of NP/molecules in order to produce Coulomb staircase. Thus, they can be scaled using industrial fabrication methods. Graphene's ability to bridge defects in the SAM and conform to dominant QDs is key to producing local electronic behaviour in devices containing thousands of chemically tunable electronic building blocks. The Coulomb-staircase profiles could be made more similar by narrowing the QD size distribution and flattening the bottom electrode topography; however, since a wide size distribution and variable topography may have the beneficial effect of reducing the number of active QDs in a junction, homogenising these could destroy the staircase behaviour. Indeed, there may be a sweet spot in between these competing effects. The use

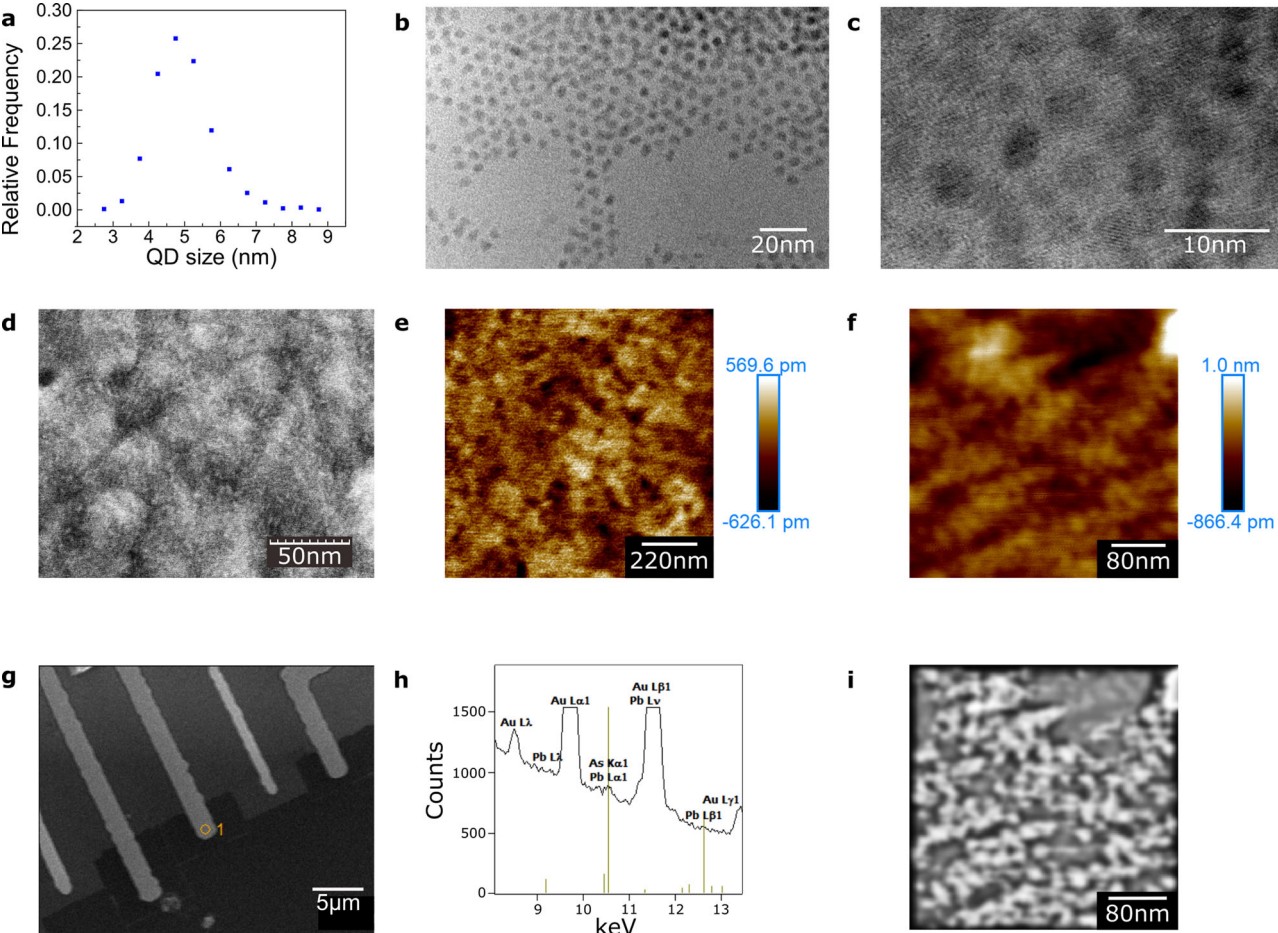

**Fig. 4 Characterisation of quantum dots. a** Normalised distribution of average widths of >1000 PbS QDs measured from TEM data taken after drop-casting QDs on to a TEM grid. $\mu = 5.0$ nm, $\sigma = 0.8$ nm. **b** TEM image showing some of the QDs after drop-casting. **c** Calibrated image where the QD crystal structures can be seen. **d, e, f** Packing density of QDs assessed by replicating the self-assembly on a quartz-crystal microbalance (QCM) sample. This shows a C6S2 number density ~5.19 molecules/nm$^2$ and a QD number density ~$5.0 \times 10^{-3}$ nm$^{-2}$: the former is consistent with recorded values for a saturated alkanethiol SAM on Au[49]. The QD number density corresponds to an average QD centre-to-centre distance ~14.1 nm, and a surface-to-surface distance ~9.2 nm. **d** SEM micrograph showing ~50 nm Au grains with QDs assembled on top. **e, f** Self-assembly resolved using a contact-mode AFM on a template-stripped Au substrate for reduced surface roughness. **g, h** Energy-dispersive X-ray spectroscopy to identify elements in different device areas. Sections where QDs are expected (Au/QD/SLG) compared with areas where they are not (Au/SLG). The PbL$\alpha$1:AuL$\alpha$1 peak-count ratio, containing the PbL$\alpha$1 line at 10.5515 keV and the AuL$\alpha$1 line at 9.7133 keV, is measured many times and averaged for comparison. The ratio is 0.027 where QDs are present and 0.0034 where they are not. These numbers are compatible with the presence of a PbS SAM of QDs. **g** SEM micrograph of the region containing Au, QD and SLG, showing with a circle the Au/QD/SLG point at which EDX is carried out. **h** Corresponding EDX data showing PbL$\alpha$1 and AuL$\alpha$1 lines. **i** AFM image in (**f**) with low frequencies removed, and 2d Gaussian smoothing applied. Counting the peaks gives a NP density ~$1.7 \times 10^{-3}$ nm$^{-2}$, three times smaller than that calculated from the QCM. This is an underestimate, as many NPs are probably not visible in this filtered picture. Even with this spacing, the voids between the irregularly arranged NPs are small enough that they should be bridged by SLG.

of layered materials as electrodes in molecular/NP SAMs paves the way to harnessing the single-electron effects in molecular/NP systems for memories, switches and sensors.

## Methods

**Device fabrication**. A 3 cm$^2$ piece of SiO$_2$/Si is used to produce 36 samples at once. Each contains 39 devices in its 160 $\mu$m × 30 $\mu$m central region, resulting in 1404 devices per batch. In principle, the entire process can be scaled up to larger dimensions.

The first stage is vacuum evaporation of the central Au regions. This produces all 1404 device electrodes, seen as vertical lines on a single sample in Fig. 1b, and three additional horizontal electrodes per sample for grounding. The smallest device, with a width ~0.8 $\mu$m, can be seen centrally on the top row. Similar processing is used to evaporate an outer region of electrodes that connect the central devices electrodes to bond pads on the perimeter of each sample.

In order to selectively chemisorb the C6S2 and assemble the QDs, OL is used (Shipley S1813 photoresist) to create a deposition window over the Au electrodes' tips. The C6S2 and QD assemblies take place in an inert nitrogen environment to prevent QD oxidation. The samples are placed in a C6S2-anhydrous isopropyl alcohol (IPA) mixture with a concentration ~1 mmol/L for 24 h. Once removed, IPA is deposited, left for 10 s, and spun off the samples at 2000 rpm three times to remove loose C6S2 molecules. Immediately following this, the substrate is placed in a 1 mg/L colloidal suspension of PbS QDs dispersed in octane for another 24 h. After the QD assembly is finished, clean octane (without QDs) is deposited on the samples, left for 10 s and spun off at 2000 rpm three times to remove loose QDs. Immersion in a large beaker of acetone for 10 min removes the remaining polymer. This is replaced with IPA. Whilst remaining in the inert environment, the samples are lifted into a PMMA/SLG membrane that is floating in DI water.

SLG is grown by chemical vapour deposition on 35 $\mu$m Cu[53]. The as-grown film is characterised by Raman spectroscopy at 514nm with a Renishaw InVia spectrometer equipped with a 50 × objective, Fig. 5, with Cu photoluminescence

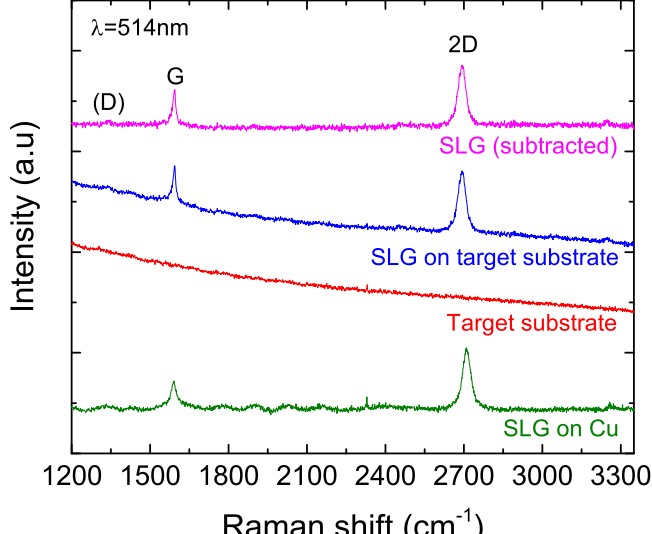

**Fig. 5 Raman spectra at 514 nm.** Green: SLG on Cu. Red: Au/QD-SAM. Blue: SLG on Au/QD-SAM. Magenta: Spectrum of SLG on Au/QD-SAM after subtracting the spectrum without the SLG.

removed[54]. The 2D peak ~ 2710 cm$^{-1}$ is a single Lorentzian, a fingerprint of SLG[55]. The G peak at ~ 1591 cm$^{-1}$ has full width at half maximum (FWHM) ~25 cm$^{-1}$, while the 2D peak has FWHM(2D) ~36 cm$^{-1}$. The 2D to G intensity and area ratios are I(2D)/I(G) ~ 2.2 and A(2D)/A(G) ~ 3.8.

To transfer SLG onto the target substrate, a PMMA layer is spin-coated on the SLG/Cu surface and then the whole PMMA+SLG/Cu stack is placed in ammonium persulphate or iron chloride for Cu etching[44]. The remaining membrane is moved into DI water for cleaning APS residuals. Samples are then lifted into the floating PMMA/SLG membrane. The transferred SLG is again characterised by Raman spectroscopy. The target substrate has a background luminescence (red line). When SLG is transferred, the background signal adds to the SLG spectrum (blue line). The D peak is negligible, indicating that the transfer process has not damaged the SLG. The positions of the G and 2D peaks are ~ 1592 cm$^{-1}$ and ~ 2692 cm$^{-1}$, respectively, with FWHM(G) ~ 16 cm$^{-1}$ and FWHM(2D) ~ 35 cm$^{-1}$, I(2D)/I(G) ~ 1.6 and A(2D)/A(G) ~ 4.2, indicative of ~ 300 meV doping[56].

The presence of the QD SAM is confirmed by AFM combined with mechanical scraping (Fig. 6), a QCM and energy-dispersive X-ray (EDX) (Fig. 4).

In both OL and EBL device fabrication the PMMA is removed from the SLG with acetone, followed by rinsing with IPA. For OL devices, optical resist is spun on SLG, and the entire area outside the central region is exposed and developed. This leaves a rectangle of resist over the central region covering just the tips of the Au electrodes. The positions of the tips are staggered across the likely positions of this rectangle's edges, caused by alignment error, to minimise device areas on each sample after the exposed SLG is etched by oxygen plasma in a reactive-ion etching (RIE) machine (20 s at 10 W and 75 mTorr). The remaining SLG only contacts the Au electrode tips, where the C6S2 SAM and QDs are assembled, and the grounding electrodes.

For EBL devices, four thicknesses (40, 50, 60, 100 nm) of 950,000 molecular-weight PMMA in anisole (1:1) are spun on the SLG. This is done to capture the minimum EBL resolution when patterning the SLG top electrodes as these define device sizes. Deep UV lithography is used to remove all but a PMMA rectangle over the samples' central regions much like on the OL samples, the difference being that these can be subsequently patterned with EBL. The PMMA development is in IPA:methyl isobutyl ketone:methyl ethyl ketone 15:5:1 for 5–10 s at 21 °C. The remaining PMMA is then patterned with EBL and the SLG is etched with RIE to create the smallest device areas in the dataset.

In both sample designs, final device areas are measured with either an SEM or an optical microscope. 18 devices of the 39 on each sample are selected for wire-bonding along with an additional 2 grounding electrodes. Control samples are fabricated with identical EBL and OL procedures but omitting the QD assembly.

Each sample package is attached to a dipstick and immersed in liquid helium to reduce the temperature to 4 K. I–V measurements are taken in triangle bias sweeps using a source-measure unit (Keithley 236) with a current resolution ~ 0.1 pA. The first sweep starts at 0 V, and is increased to ~ 0.1 V and then swept down to an equivalent negative voltage and finally back to zero. The magnitudes of these triangle bias sweeps are then increased in increments of 0.1–0.5 V until the extremely high electric field causes the device's behaviour to become erratic.

**Molecular barriers.** The C6S2 hexanedithiol molecules in the dense SAM consist of six carbon units (C6) and two thiol terminal groups (S2) for anchoring to both QDs and Au. The inclusion of more than one thiol anchor allows for an alternative linking scenario where C6S2 is attached as a chelating ligand to the Au surface. This may preclude a direct covalent linkage to the QD, but our assembly process is designed to lower the occurrence of this binding mode. The weakly bound oleic-acid capping ligands prevent QD agglomeration in the octane. During assembly these are displaced locally through an exchange process to form chemical bonds between QDs and C6S2 molecules[57,58]. This immobilises the QDs and allows for the removal of any excess QDs, ensuring a monolayer (Fig. 4).

Using molecular modelling[59], we estimate the S-to-S length of C6S2 to be 0.94 nm. Alkanethiolate SAMs assembled on Au are reported to have a typical tilt angle of 28° with respect to the surface normal[49]. This allows us to estimate the film thickness to be ~ 0.83 nm. When C6S2 molecules replace the oleic acid ligands that surround the PbS QD, free S atoms at the surface of the C6S2 SAM bond to the QD's surface anchoring it to the substrate, and allowing the formation of a QD SAM. The oleic acid itself provides the other tunnel barrier in the junction separating the QD and the SLG. In 6 nm PbSe QDs, it has been reported that oleic-acid molecules conform in such a way as to produce a capping layer of thickness ~ 1 nm, despite having an isolated length ~ 2 nm[60]. These ligands behave similarly with PbS QDs, so the asymmetric-barrier condition should often be fulfilled.

**AFM with mechanical scraping.** The selective formation of the QD SAM is confirmed by a mechanical cleaning process that uses an AFM tip to sweep away the assembly in specific device regions, so a height difference can be measured[61–63] (Fig. 6). A small area of the SAM is scanned repeatedly (16 times) in contact mode at high force (30 nN), then scanned over a larger area at low normal force (2 nN). A clean 'window' is observable in the small high-force-scanned region where the SAM is scratched away so its height can be determined (5.1 ± 0.9 nm). This matches the TEM measurements (Fig. 4) of the diameters of the NPs. Scanning is repeated in areas where no SAM is expected to form, due to photoresist protection during assembly. A height difference of 1.3 ± 0.4 nm is measured (21 sweeps).

**Quartz crystal microbalance.** A QCM is used to measure the surface functionalisation of the Au electrodes. When an alternating voltage is applied to two electrodes of known geometry sandwiching a quartz plate, the current response has a resonant frequency. This is shifted by the deposition of a film on the electrodes' surfaces and can be measured by a frequency counter. The linear relation between the observed frequency shift and mass deposited is given by the Sauerbrey equation[64]:

$$\Delta f = -2\frac{f_0^2 \Delta m}{A\sqrt{\mu_q \rho_q}}, \qquad (1)$$

where $\Delta f$ is the frequency shift, $f_0$ the resonant frequency, $\Delta m$ the mass deposited, $A$ the area of the electrode on the QCM, $\mu_q$ the shear modulus of quartz, and $\rho_q$ the density of quartz. In our system, with a 10 MHz QCM, the relation between frequency shift and mass per unit area is:

$$\Delta f = -4.5 \times 10^{13} \,\mathrm{Hz/(ng/nm^2)} \frac{\Delta m}{A}. \qquad (2)$$

The frequency shift after the self-assembly of C6S2 on the QCM's Au electrodes is ~ − 58.3 Hz, corresponding to a packing density ~ 5.19 molecules/nm$^2$. After the QD assembly, this is ~ − 135.6 Hz, representing a mass-per-unit-area gain of ~ 3.0 × 10$^{-12}$ ng nm$^{-2}$. Using the size distribution obtained through TEM, and assuming an OA ligand packing density ~ 4 nm$^{-2}$ on the QDs' surface, we estimate the average QD mass ~ 6.4 × 10$^{-10}$ ng. From this we deduce a coverage ~ 5.0 × 10$^{-3}$ nm$^{-2}$ and an average centre-to-centre distance between neighbouring QDs ~ 14.1 nm, and 9.2 nm surface-to-surface. The poor mechanical coupling between the heavy QDs and the substrate means that QCM may underestimate the QD number density so the gaps may be smaller than this.

**Template stripping.** Au substrates are prepared using standard template-stripping recipes[65,66]. Thermally evaporated Au on SiO$_2$/Si is transferred onto a second SiO$_2$/Si wafer using epoxy to create an Au surface with roughness ~ 0.15 ± 0.2 nm. The self-assembly recipe is used to create the QD SAM. Contact-mode AFM with a NuNano Scout 70 tip is then used to resolve the QDs (Fig. 4e,f). These appear wider than expected due to a common artefact of AFM imaging which exaggerates the lateral dimensions of nanoscale protuberances[67].

## Data availability
Data associated with this work are available[68].

## Code availability
Scripts for analysing the data associated with this work are available[68].

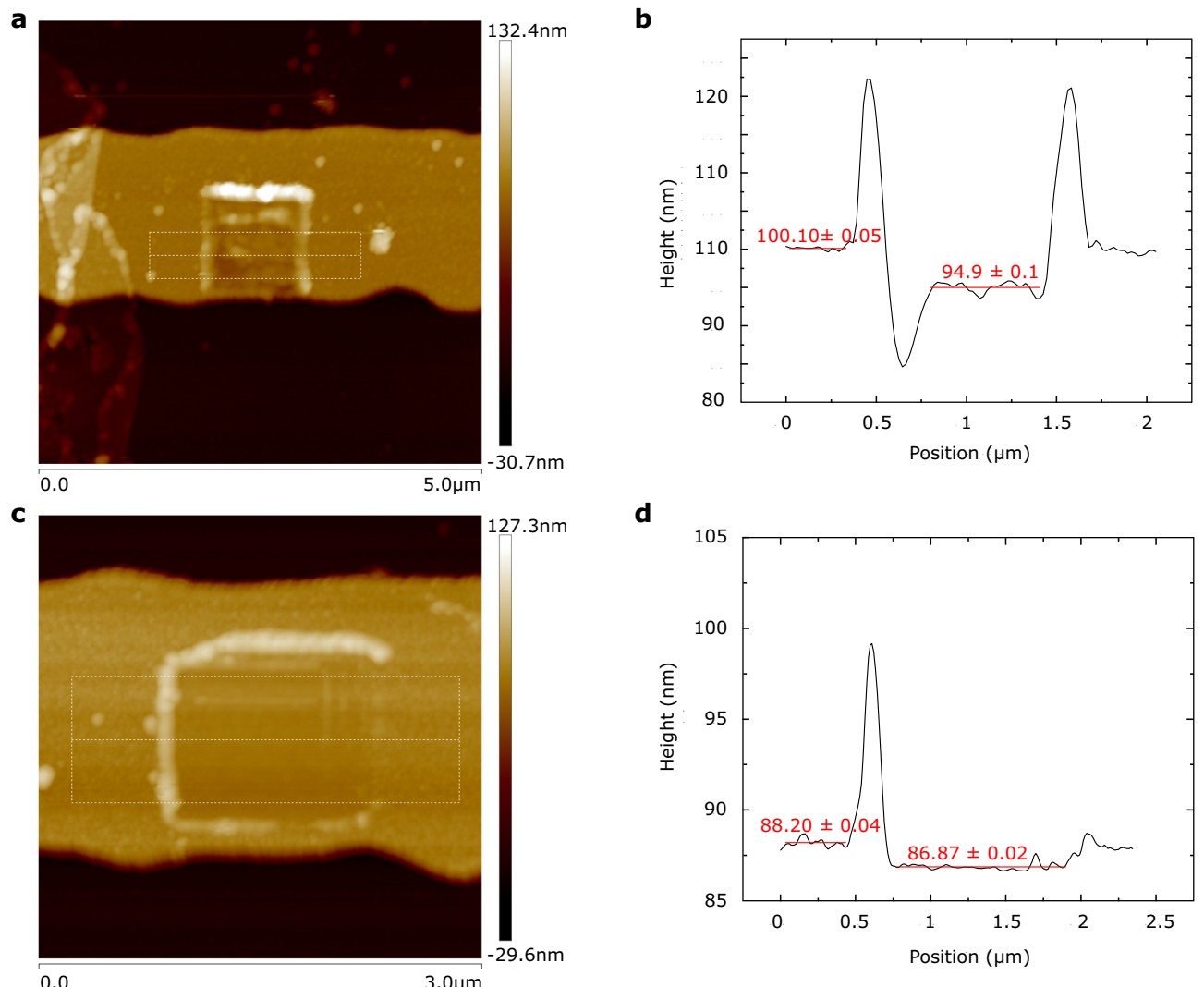

**Fig. 6 AFM imaging of device regions verifying the formation of a QD monolayer. a–b** Height profile across an Au electrode where the QD SAM is first assembled and subsequently scraped away mechanically using an AFM tip. Measured height difference ~ 5.2 nm. **c–d** Height profile across an Au electrode where the QD SAM is not assembled, due to photoresist protection, but with the area still scraped mechanically with the AFM tip. Measured height difference ~1.3 nm.

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

## Acknowledgements

We thank Yiqing Jin, Antonio Lombardo and Angelo Di Bernardo for experimental help in the early stages of the project, James Xiao for discussion regarding quantum-dot chemistry, Xintai Wang for SAM characterisation, and Richard Langford and Jon J. Rickard for electron-microscopy support. This work was supported by UK EPSRC grants EP/P027172/1, EP/K01711X/1, EP/K017144/1, EP/N010345 /1, EP/K016636/1, EP/N010345/1, EP/L016087/1, the EU Graphene and Quantum Flagships, ERC grants Hetero2D and Minergrace.

## Author contributions

Conception and supervision of the project: C.J.B.F.; Development of the fabrication and $I - V$ measurement schemes: J.M.F.; Device fabrication: J.M.F. with help from HPAGA; Detailed $I - V$ measurements: J.M.F., H.P.A.G.A. and L.F.L.E.; Data-analysis tools: H.P. A.G.A., C.J.B.F., and J.M.F.; Analysis of results and interpretation: J.M.F./H.P.A.G.A. and C.J.B.F.; Characterisation of SAM by QCM: H.P.A.G.A.; Atomic force microscopy and ultrasonic force microscopy: A.R. and B.J.R. in discussion with H.P.A.G.A .and J.M.F.; Paper preparation: J.M.F., H.P.A.G.A., C.J.B.F. and A.C.F.; Graphene growth, characterisation and transfer: P.R.K., U.S., D.D.F., S.H. and A.C.F.; Quantum-dot growth and characterisation: B.E. and M.B.; Electron-beam lithography: J.P.G.

## Competing interests

The authors declare no competing interests.
