## [Peer Review File · Nature Communications]

REVIEWER COMMENTS

Reviewer #1 (Remarks to the Author):

The manuscript "High-yield parallel fabrication of quantum-dot monolayer single-electron devices displaying Coulomb staircase, contacted by graphene" by Christopher J.B. Ford and co-workers. Considers the fabrication and study of single electron transistors made from monolayers of quantum dots coupled to graphene electrodes.

While not of molecular dimensions, It is a respectable achievement to reach 70% device yield, since the device yield in this new structural arrangement is (in my view) the main progress here, I suggest the authors to revise how they present this yield, several different yields are quoted in the manuscript which adds to some confusion. The device yields and functional device yields are difficult to understand for the reader, e.g. it is stated in one place that "80% of our devices do not short electrically and half of these show coulomb staircase" earlier it is stated that "Here, we present a SLG-covered SAM of NPs that produces functioning single-electron transport (SET) devices displaying a Coulomb staircase in their I-V characteristics with a yield of $87 \pm 13\%$ " the different yields and statements are a bit confusing.

The manuscript is not so easy to follow and reads to some extent like an extended lab report. The difference in performance between optical lithography and EBL devices are in my view only interesting from a more technically interested reader, and I suggest to take one of them out of the main manuscript, to make a more concise and to the point presentation.

In summary In my view, the manuscript is publishable, but the presentation needs to be improved as noted.

Additional comments

- 1) In the introduction, the authors state "How- ever, the severe difficulty of contacting such small objects with a wafer-scale process has thus far prevented the integration of nanocrystals and molecules into conventional microelectronic circuitry. " while I agree with this statement, several reports in recent 3-4 years have tried to address that, some with quite good results, for much smaller device dimensions see e.g. <https://www.nature.com/articles/s41565-019-0533-8> and Nature Communications volume 9, Article number: 3433 (2018) and Hihath <https://doi.org/10.1002/adfm.202000615> there are quite some more recent examples, I suggest that the authors integrate somewhat better recent works in order to put their work in context of state of the art. (most of given examples (ref 15-21, are 5-15 years old) also, please consider the similarities with Wiel nature nano volume 10, pages1048-1052(2015)
- 2) The authors state that they have a SAM (= self-assembled monolayer) of quantum dots. The drawing in fig 1a shows 6 layers of quantum dots, which in my view is not a monolayer. I thus suggest to change the term SAM to something else, since the device is not a monolayer (while the active part might be).

Reviewer #2 (Remarks to the Author):

Review of article: 'High-yield parallel fabrication of quantum-dot monolayer single-electron device displaying Coulomb staircase, contacted by graphene' by Fruhman et. al.

The major claim of the paper is the demonstration of a scalable technique for contacting small molecules and nanocrystals that is compatible with conventional microelectronic fabrication thereby allowing hybrid electronics. The team supports this claim by using a combination of conventional lithography, self-assembly for PBS quantum dots, and wet transfer of single layer graphene to build 1400 single-electron transport devices per batch. Each device itself consists of thousands of gold/PBS quantum dot/single layer graphene units in parallel and the single-electron transport behavior is confirmed with observation of Coulomb staircase in the I-V characteristics of these devices with a yield of $> 70\%$.

The claims are novel, and work carried out by the authors is of great interest as scaling up nanoscale devices while preserving unique electronic phenomena has always been challenging. There is a lot of published work with Coulomb staircase devices, but these are limited to nanogap or nano electrode devices which contact a single or small number of molecules, a structure which is not scalable. Going to larger number of molecules allows scalable and repeatable fabrication however the average of contributions from individual molecules dilutes the Coulomb staircase. The work under review utilizes a surface assembled monolayer of quantum dots over a wide area which allows easy placement of the top graphene contact. The conformity of graphene layer allows for a global contact over disperse quantum dot widths in the SAM but still allows predominantly local effects such that only a few quantum dots actively participate in transport and hence preserve the Coulomb staircase behavior. The interesting and surprising and promising aspect of these devices is the ability to preserve the Coulomb blockade behavior over multiple I-V sweeps and the authors claim only minor variation even after >100 sweeps.

While the stated yield of Coulomb blockade devices of > 70% is unparalleled (as rightly pointed out in a detailed comparison table Table S1), the bigger concern one has is the repeatability or uniformity of the responses across devices not just in one device - that is if one makes hundreds of these devices with similar dimensions, will the response of each one would be identical? The step heights for the devices even in the same batch and for same device area vary by $\sim 1000\times$ (Fig 4a). This hints that many of the mechanisms which contribute to the Coulomb staircase are not controlled. The authors also point out that the tunnel barrier length especially at the QD-graphene junction can vary. This could be due to the variation in QD size distribution and/or residue on graphene during wet processing (also wet processing of graphene as a scalable process is questionable). This raises concerns and it is important to understand if the process is controllable enough and repeatable over batches or it is just good fortune. The authors state that each fab run produces 36 chips with 39 devices each. Have the authors studied the distribution of staircase parameters across the 36 chips? What is the variation they observe across the 36 chips and not just across 18 out of 39 devices which are wire bonded in one chip?

It is also not clear from the manuscript whether there are devices with the same length (L) and width (W) on the same chip, what is their count and how do they compare? A follow up experiment would be to make multiple identical devices (same L and W and hence same area and edge effects) and see how the distribution of step height and step width of Coulomb staircase vary. While not difficult, this follow up study would require time of course. Comparison of identical devices from different chips would also be interesting to study, if that data is already available.

Ten devices with identical dimensions and identical fabrication process showing identical electrical behavior is far more valuable and promising for large scale microfabrication technology than thousand devices with 70% yield and varying electrical characteristics. Frankly, one of the challenges of nanoscale electronics is the difficulty of achieving tight control on the spread of electrical parameters as one moves towards batch fabrication. Yield may be 100% but if variation amongst identical devices is high, it cannot make its way into integrated circuits like memories, switches, sensors, or thermoelectric generators as envisioned by the authors.

- Are the claims novel?

Yes.

- Will the paper be of interest to others in the field?

Yes

- Will the paper influence thinking in the field?

Maybe

- Are the claims convincing? If not, what further evidence is needed?

The authors need to provide evidence of the repeatability of the characteristics for devices which have similar area and manufactured in the same way. Just having a high yield of working devices is not a sufficient condition for adoption into integrated electronics let alone by other researchers for further investigation.

- Are the claims appropriately discussed in the context of previous literature?

Yes. The acknowledgement of past work is commendable and is extensive.

- If the manuscript is unacceptable in its present form, does the study seem sufficiently promising that the authors should be encouraged to consider a resubmission in the future?

Yes

- Is the manuscript clearly written?

Yes.

- Could the manuscript be shortened to aid communication of the most important findings?

Maybe but not an issue.

Response to reviewers

Reviewer #1:

1) The manuscript “High-yield parallel fabrication of quantum-dot monolayer single-electron devices displaying Coulomb staircase, contacted by graphene” by Christopher J.B. Ford and co-workers. Considers the fabrication and study of single electron transistors made from monolayers of quantum dots coupled to graphene electrodes. While not of molecular dimensions, it is a respectable achievement to reach 70% device yield, since the device yield in this new structural arrangement is (in my view) the main progress here, I suggest the authors to revise how they present this yield, several different yields are quoted in the manuscript which adds to some confusion. The device yields and functional device yields are difficult to understand for the reader, e.g. it is stated in one place that “80% of our devices do not short electrically and half of these show coulomb staircase” earlier it is stated that “Here, we present a SLG-covered SAM of NPs that produces functioning single-electron transport (SET) devices displaying a Coulomb staircase in their I-V characteristics with a yield of $87 \pm 13\%$ ” the different yields and statements are a bit confusing.

We thank the reviewer for pointing out the seeming inconsistency in how the yield is reported in various parts of the paper. We have updated the manuscript to clarify that the optimised yield of $87 \pm 13\%$ was achieved for devices fabricated with areas less than $2 \mu\text{m}^2$ (see line 41 on page 2 in the manuscript).

2) The difference in performance between optical lithography and EBL devices are in my view only interesting from a more technically interested reader, and I suggest to take one of them out of the main manuscript, to make a more concise and to the point presentation.

We acknowledge the suggestion from reviewer 1 that we present only one type of lithography but we would like to point out that the EBL and OL devices differ in a number of notable ways, not just in their area ranges and fabrication processes. Most noteworthy is the size of the open part of the etched graphene, relative to the device area. The EBL fabrication method produces devices that have a much larger open etched perimeter, which we think results in less shorting, due to the SLG being flatter. We feel that these devices are interesting in their own right and that since so many of our cleanest Coulomb-staircase measurements are from EBL devices and they are a significant part of the data, contributing to the error bar on the yield number, it would be misleading not to include them.

3) In summary In my view, the manuscript is publishable, but the presentation needs to be improved as noted.

We hope that the presentation is now acceptable.

4) Additional comments

1) In the introduction, the authors state “However, the severe difficulty of contacting such small objects with a wafer-scale process has thus far prevented the integration of nanocrystals and molecules into conventional microelectronic circuitry.” while I agree with this statement, several reports in recent 3-4 years have tried to address that, some with quite good results, for much smaller device dimensions see e.g. [<https://www.nature.com/articles/s41565-019-0533-8>] and Nature Communications volume 9, Article number: 3433 (2018) and Hihath [<https://doi.org/10.1002/adfm.202000615>] there are quite some more recent examples, I suggest that the authors integrate somewhat better recent works in order to put their work in context of state of the art. (most of given examples (ref 15-21, are 5-15 years old) also, please consider the similarities with Wiel nature nano volume 10, pages1048-1052(2015)

2) The authors state that they have a SAM (= self-assembled monolayer) of quantum dots. The drawing in fig 1a shows 6 layers of quantum dots, which in my view is not a monolayer. I thus suggest to change the term SAM to something else, since the device is not a monolayer (while the active part might be).

1) We have included the references mentioned and some other more recent references.

2) We have edited the labels in Fig. 1a to show more clearly that the SLG is draped over the edge of the Au finger, which we happened to show as 6

QDs high, but there is still expected to be a monolayer of QDs between the graphene and the Au. We have evidence for there being a single monolayer on the upper Au surface from various imaging techniques and the quartz crystal microbalance (as described in the Extended Data).

Reviewer #2:

5) Review of article: ‘High-yield parallel fabrication of quantum-dot monolayer single-electron device displaying Coulomb staircase, contacted by graphene’ by Fruhman et. al. The major claim of the paper is the demonstration of a scalable technique for contacting small molecules and nanocrystals that is compatible with conventional microelectronic fabrication thereby allowing hybrid electronics. The team supports this claim by using a combination of conventional lithography, self-assembly for PBS quantum dots, and wet transfer of single layer graphene to build 1400 single-electron transport devices per batch. Each device itself consists of thousands of gold/PBS quantum dot/single layer graphene units in parallel and the single-electron transport behavior is confirmed with observation of Coulomb staircase in the I-V characteristics of these devices with a yield of > 70%. The claims are novel, and work carried out by the authors is of great interest as scaling up nanoscale devices while preserving unique electronic phenomena has always been challenging. There is a lot of published work with Coulomb staircase devices, but these are limited to nanogap or nano electrode devices which contact a single or small number of molecules, a structure which is not scalable. Going to larger number of molecules allows scalable and repeatable fabrication however the average of contributions from individual molecules dilutes the Coulomb staircase. The work under review utilizes a surface assembled monolayer of quantum dots over a wide area which allows easy placement of the top graphene contact. The conformity of graphene layer allows for a global contact over disperse quantum dot widths in the SAM but still allows predominantly local effects such that only a few quantum dots actively participate in transport and hence preserve the Coulomb staircase behavior. The interesting and surprising and promising aspect of these devices is the ability to preserve the Coulomb blockade behavior over multiple I-V sweeps and the authors claim only minor variation even after > 100 sweeps.

We thank the referee for recognising that preserving ‘local effects’ in a scalable molecular device, is ‘novel’ and ‘of great interest’ and that until now all research has been ‘limited to nanogap or nano electrode devices which contact a single or small number of molecules, a structure which is not scalable’, and also for noting that the ‘interesting and surprising and promising aspect of these devices is the ability to preserve the Coulomb blockade behaviour over multiple I-V sweeps’.

6) While the stated yield of Coulomb blockade devices of $> 70\%$ is unparalleled (as rightly pointed out in a detailed comparison table Table S1), the bigger concern one has is the repeatability or uniformity of the responses across devices not just in one device - that is if one makes hundreds of these devices with similar dimensions, will the response of each one would be identical? The step heights for the devices even in the same batch and for same device area vary by $\sim 1000\times$ (Fig 4a). This hints that many of the mechanisms which contribute to the Coulomb staircase are not controlled. The authors also point out that the tunnel barrier length especially at the QD-graphene junction can vary. This could be due to the variation in QD size distribution and/or residue on graphene during wet processing (also wet processing of graphene as a scalable process is questionable). This raises concerns and it is important to understand if the process is controllable enough and repeatable over batches or it is just good fortune. The authors state that each fab run produces 36 chips with 39 devices each. Have the authors studied the distribution of staircase parameters across the 36 chips? What is the variation they observe across the 36 chips and not just across 18 out of 39 devices which are wire bonded in one chip?

We note the referee describes our yield as ‘unparalleled’, and thank them for the suggestions here. We have indeed looked at staircase parameters across many chips in each batch of 36 chips (fabricated as one piece of material) and across multiple fab runs, and all the data are presented here in the scatter plots and histograms. Figure 4a shows that when devices are separated by batch, device area does indeed predict step height. There is no separation at the chip level in this figure and we have confirmed separately that any observed patterns are not chip-dependent. In fact, even at the batch level (for the same lithography type) the step heights for each cluster only differ by a factor of about two, which is all the more promising as far as reproducibility goes. The only difficulty is that the step heights break up into different discrete conduction groups (discussed on page 6, lines 32–47) which implies that there are two, or for the EBL samples three, mechanisms that we are currently unable to select in advance. The existing devices always have edges as well as a central area, each of which may contribute a conduction mechanism, and we believe it is likely that improved device designs (e.g. ‘microwells’) will help by avoiding tunnelling conduction at the edges. We have added additional histograms to Fig. 4 b–d which highlight these groupings more clearly, by plotting the ratio of step height to junction area, and this shows two (or three) narrow distributions representing the diagonal lines of points in the scatter plots in Fig. 4a (with matching colours). The width of each distribution is less than an order of magnitude, which is the best achieved for any conductance measurement of sets of single molecules or molecular SAMs.

Whilst we appreciate that it is not clear how this functionality might be used in mass-fabricated devices today, we feel that the level of reproducibility observed, combined with the extremely high yield of single-electron behaviour, is completely novel and remarkable and should result in more research into how these fabrication methods might be used to produce high-yield, scaleable molecular or quantum-dot devices. In the paper’s conclusions we state that ‘the

Coulomb-staircase profiles could likely be made more similar by narrowing the QD size distribution and flattening the bottom electrode topography’.

The referee also comments that wet transfer of graphene is not scaleable, which is, of course, true, but there is no reason to believe that roll-to-roll graphene (which is rapidly becoming more practicable, see e.g. Tavakoli *et al.*, *Adv. Funct. Mater.* **30**, 2001924 (2020)) would not work just as well, once the technology is better developed.

7) It is also not clear from the manuscript whether there are devices with the same length (L) and width (W) on the same chip, what is their count and how do they compare? A follow up experiment would be to make multiple identical devices (same L and W and hence same area and edge effects) and see how the distribution of step height and step width of Coulomb staircase vary. While not difficult, this follow up study would require time of course. Comparison of identical devices from different chips would also be interesting to study, if that data is already available.

Currently, no two devices ever have identical L and W because of the lithography process that was used (page 2, line 67). However, the similarity of step heights for devices with almost identical areas can be seen in Fig. 4a, and the newly included histograms of step height/area (Fig. 4b-d) show very narrow distributions for each trend. There is a much worse correlation with perimeter (either total, or just ‘open’ or ‘closed’ edges). It is not practical to produce a set of identical devices in the near future, but by showing the dependence on area and perimeter we have gone one step further.

8) Ten devices with identical dimensions and identical fabrication process showing identical electrical behavior is far more valuable and promising for large scale microfabrication technology than thousand devices with 70% yield and varying electrical characteristics. Frankly, one of the challenges of nanoscale electronics is the difficulty of achieving tight control on the spread of electrical parameters as one moves towards batch fabrication. Yield may be 100% but if variation amongst identical devices is high, it cannot make its way into integrated circuits like memories, switches, sensors, or thermoelectric generators as envisioned by the authors.

Even for devices contacting simple molecular SAMs, the best results from the literature show a spread of a least an order of magnitude in conductance per unit area (e.g. G. Puebla-Hellmann *et al.*, *Nature*, **559**, 7713, 232, 2018). Our Coulomb staircases show that there is clear predictability of step height for a given area, with a similar spread to Coulomb-blockade devices that can never be scalable. Because of this, these single-electron effects could potentially be taken advantage of in microelectronics. We want to encourage other researchers to look for ways to make the output even more predictable. This is something we are also pursuing in our research group.

9) • Are the claims novel?

Yes.

• Will the paper be of interest to others in the field?

Yes

• Will the paper influence thinking in the field?

Maybe

• Are the claims convincing? If not, what further evidence is needed

The authors need to provide evidence of the repeatability of the characteristics for devices which have similar area and manufactured in the same way. Just having a high yield of working devices is not a sufficient condition for adoption into integrated electronics let alone by other researchers for further investigation.

As we have addressed in the detailed responses to Referee 2, we have added histograms of step height scaled by area to show that the distribution of values is narrower than an order of magnitude, and is therefore at the state of the art. To improve this will take significant further optimisation, beyond the scope of this paper, but we have shown the way to make scaleable junctions showing steps.

10) • Are the claims appropriately discussed in the context of previous literature?

Yes. The acknowledgement of past work is commendable and is extensive.

• If the manuscript is unacceptable in its present form, does the study seem sufficiently promising that the authors should be encouraged to consider a resubmission in the future?

Yes

• Is the manuscript clearly written?

Yes.

• Could the manuscript be shortened to aid communication of the most important findings?

Maybe but not an issue.

We hope the referees will find our responses clear and will now support publication of the work in *Nature Communications*.

REVIEWERS' COMMENTS

Reviewer #1 (Remarks to the Author):

The authors have done a good job in revising their manuscript, and I can now recommend publication, pending minor revision.

In the experimental part, the synthesis of the QD is not well described. What is the used method? what was the quality and source of used chemicals? the concentration noted as 1 mg/L is a quite crude way of describing it, what is the expected number of particles/L? what is the composition of the particles? and are they expected to have an oxide shell? This information is important for someone who might be interested in reproducing the work. The coating shell thickness might influence the Coulomb charging effects.

Reviewer #2 (Remarks to the Author):

While reviewing the article the first time, the major concern was the variability in the Coulomb Blockade parameters across the devices since the main claim of the manuscript was a high-yield parallel fabrication of QD single electron devices. By adding additional plots 4b-d, the authors have highlighted that the variation in Coulomb step heights with area is within a factor of 5 and acknowledge that further optimization is needed. I believe the authors have done the best they can with available devices and data; conducting another set of experiments where identical devices are fabricated and edge effects are controlled can be a follow-up effort.

The manuscript is publishable however I do believe the manuscript can be written better. Nature journals pride themselves in publishing articles that are accessible, clear, and concise. Throwing in all the data generated during the project can distract readers from the relevance and importance of the underlying study.

Response to Reviewers

Reviewer #1 (Remarks to the Author):

The authors have done a good job in revising their manuscript, and I can now recommend publication, pending minor revision.

In the experimental part, the synthesis of the QD is not well described. What is the used method? what was the quality and source of used chemicals? the concentration noted as 1 mg/L is a quite crude way of describing it, what is the expected number of particles/L? what is the composition of the particles? and are they expected to have an oxide shell? This information is important for someone who might be interested in reproducing the work. The coating shell thickness might influence the Coulomb charging effects.

A detailed description of the QD synthesis is now referenced in Gao2011 'Quantum Dot Size Dependent J-V Characteristics in Heterojunction ZnO-PbS Quantum Dot Solar Cells'.

Reviewer #2 (Remarks to the Author):

While reviewing the article the first time, the major concern was the variability in the Coulomb Blockade parameters across the devices since the main claim of the manuscript was a high-yield parallel fabrication of QD single electron devices. By adding additional plots 4b-d, the authors have highlighted that the variation in Coulomb step heights with area is within a factor of 5 and acknowledge that further optimization is needed. I believe the authors have done the best they can with available devices and data; conducting another set of experiments where identical devices are fabricated and edge effects are controlled can be a follow-up effort.

The manuscript is publishable however I do believe the manuscript can be written better. Nature journals pride themselves in publishing articles that are accessible, clear, and concise. Throwing in all the data generated during the project can distract readers from the relevance and importance of the underlying study.

The paper has been reformatted to make it clearer and more concise. Content from the main paper has been moved to the supplementary section, wording has been improved and non-essential figures have been moved out of the main paper.